# *Pdx1* Is Transcriptionally Regulated by EGR-1 during Nitric Oxide-Induced Endoderm Differentiation of Mouse Embryonic Stem Cells

**DOI:** 10.3390/ijms23073920

**Published:** 2022-04-01

**Authors:** Carmen Salguero-Aranda, Amparo Beltran-Povea, Fátima Postigo-Corrales, Ana Belén Hitos, Irene Díaz, Estefanía Caballano-Infantes, Mario F. Fraga, Abdelkrim Hmadcha, Franz Martín, Bernat Soria, Rafael Tapia-Limonchi, Francisco J. Bedoya, Juan R. Tejedo, Gladys M. Cahuana

**Affiliations:** 1Department of Pathology, Institute of Biomedicine of Seville (IBiS), Virgen del Rocio University Hospital, CSIC-University of Seville, 41013 Seville, Spain; 2Spanish Biomedical Research Network Centre in Oncology, CIBERONC of the Carlos III Health Institute (ISCIII), 28029 Madrid, Spain; 3Department of Normal and Pathological Cytology and Histology, School of Medicine, University of Seville, 41004 Seville, Spain; 4Department of Molecular Biology and Biochemical Engineering, Universidad Pablo de Olavide, 41013 Seville, Spain; mabelpov@gmail.com (A.B.-P.); fatimapostigocorrales@gmail.com (F.P.-C.); estefania.caballano@cabimer.es (E.C.-I.); khmadcha@upo.es (A.H.); fmarber@upo.es (F.M.); fbedber@upo.es (F.J.B.); jrtejhua@upo.es (J.R.T.); 5Biomedical Research Network for Diabetes and Related Metabolic Diseases-CIBERDEM of the Carlos III Health Institute (ISCIII), 08036 Madrid, Spain; ahitos@iib.uam.es (A.B.H.); irene.diaz@cabimer.es (I.D.); bernat.soria@umh.es (B.S.); 6Department of Regeneration and Cell Therapy Andalusian, Center for Molecular Biology and Regenerative Medicine (CABIMER), University of Pablo de Olavide-University of Seville-CSIC, 41013 Seville, Spain; 7Nanomaterials and Nanotechnology Research Center (CINN-CSIC), Cancer Epigenetics and Nanomedicine Laboratory, 33940 El Entrego, Spain; mffraga@cinn.es; 8Health Research Institute of Asturias (ISPA), 33011 Oviedo, Spain; 9Institute of Oncology of Asturias (IUOPA), University of Oviedo, 33006 Oviedo, Spain; 10Rare Diseases CIBER (CIBERER) of the Carlos III Health Institute (ISCIII), 28029 Madrid, Spain; 11Department of Biotechnology, University of Alicante, 03690 Alicante, Spain; 12Health Research Institute-ISABIAL Dr Balmis University Hospital and Institute of Bioengineering, University Miguel Hernández de Elche, 03010 Alicante, Spain; 13Tropical Disease Institute, Universidad Nacional Toribio Rodríguez de Mendoza, Amazonas 01001, Peru; rafael.tapia@untrm.edu.pe

**Keywords:** EGR-1, *Pdx1*, nitric oxide, mESCs, endoderm differentiation

## Abstract

The transcription factor, early growth response-1 (EGR-1), is involved in the regulation of cell differentiation, proliferation, and apoptosis in response to different stimuli. EGR-1 is described to be involved in pancreatic endoderm differentiation, but the regulatory mechanisms controlling its action are not fully elucidated. Our previous investigation reported that exposure of mouse embryonic stem cells (mESCs) to the chemical nitric oxide (NO) donor diethylenetriamine nitric oxide adduct (DETA-NO) induces the expression of early differentiation genes such as pancreatic and duodenal homeobox 1 (*Pdx1*). We have also evidenced that *Pdx1* expression is associated with the release of polycomb repressive complex 2 (PRC2) and P300 from the *Pdx1* promoter; these events were accompanied by epigenetic changes to histones and site-specific changes in the DNA methylation. Here, we investigate the role of EGR-1 on *Pdx1* regulation in mESCs. This study reveals that EGR-1 plays a negative role in *Pdx1* expression and shows that the binding capacity of EGR-1 to the *Pdx1* promoter depends on the methylation level of its DNA binding site and its acetylation state. These results suggest that targeting EGR-1 at early differentiation stages might be relevant for directing pluripotent cells into *Pdx1*-dependent cell lineages.

## 1. Introduction

Early growth response-1 (EGR-1) is a zinc-finger transcription factor that regulates the expression of numerous differentiation and growth genes in response to environmental signals. EGR-1 expression is rapidly induced through stimulation with many environmental signals including growth factors, hormones, neurotransmitters, cellular stress, injury, mitogens, and cytokines [1,2]. EGR-1 activity is modulated in part through interaction with its co-repressors NGFI-A binding protein 1 (NAB1) and NGFI-A binding protein 2 (NAB2). It is known that EGR-1 induces the expression of *Nab2*, thereby preventing permanent transactivation of EGR-1 target genes [3].

EGR-1 is reported to be involved in a multitude of signaling pathways, standing out in the response to oxidative stress and apoptosis [4] and in reactive oxygen species (ROS)-mediated signaling and inflammation [5,6,7,8,9,10,11]. Additionally, in several types of tumor cells, EGR-1 exhibits suppressor gene activity [12,13,14]. Furthermore, EGR-1 is known to play a key role in cell differentiation. Thus, it has been described that EGR-1 can promote differentiation of bovine skeletal muscle-derived satellite cells [15,16] and osteogenic differentiation in human ligament periodontal stem cells [17]. It has also been described that EGR-1 expression favors white adipocyte differentiation and represses white adipose tissue browning [18].

Interestingly, EGR-1 has recently been reported as a critical factor to promote β cell identity, maintain β cell characteristics, and repress non-β cell programs [19]. This study describes how the loss of EGR-1 uncouples metabolic stress from the transcriptional cascade essential for the β cell compensatory response, which underscores EGR-1 as a critical factor in the pathogenesis of pancreatic islet failure. In fact, it was previously reported that EGR-1 contributes to the glucose responsiveness and function of pancreatic β cells through the regulation of insulin and pancreatic and duodenal homeobox 1 (*Pdx1*) expression [20,21]. These studies revealed the mechanisms by which EGR-1 regulates the expression of *Pdx1* in pancreatic β cells. They identified EGR-1 recognition sites within the *Pdx1* promoter (regulatory areas referred as III and IV by Gerrish et al. [22]) that contribute to its responsiveness to regulation by EGR-1. It has been proposed that the suppression of *Egr-1* expression may be an important factor for the differentiation of embryonic stem cells (ESCs) into pancreatic cells [23]. In the same way, it has recently been reported that the inhibition of *Egr-1* in the early phase induces the differentiation of pluripotent stem cells (PSCs) into pancreatic endoderm and insulin-producing cells; in contrast, the inhibition of *Egr-1* in the late phase suppresses the differentiation process [24]. These results suggest that *Egr-1* may play a dual role in β cell development, playing a repressive role in early development and then engaging in positive activity at a later stage. However, the mechanisms underlying *Egr-1* and β cell development at early stages are not fully elucidated.

We previously reported that exposure to high concentrations of the chemical nitric oxide (NO) donor diethylenetriamine nitric oxide adduct (DETA-NO) causes ESC differentiation by downregulating pluripotency genes *Nanog* and *Oct4*, and upregulating differentiation genes such as *Pdx1* [25]. We later elucidated the repressor role of polycomb repressor complex 2 (PRC2) and histone acetyltransferase P300 (P300) on *Pdx1* gene expression. Additionally, epigenetic study of the *Pdx1* promoter reveals that NO induces changes in the methylation pattern and leads to a shift in the balance of H3K27me3 and H3K4me3 occupancy [26]. In the present study, we investigate the involvement of EGR-1 in the regulation of *Pdx1* expression in mouse ESCs (mESCs) after exposure to DETA-NO. The results show that EGR-1 represses *Pdx1* gene expression in mESCs. Mechanistically, we propose that DETA-NO modifies the methylation degree of the EGR-1 binding site to the *Pdx1* promoter and EGR-1 acetylation state, and consequently, reduces the EGR-1 binding capacity to the *Pdx1* promoter, thus allowing *Pdx1* expression.

## 2. Results

### 2.1. EGR-1 Binds to Pdx1 Promoter in mESCs

We previously described how *Pdx1* expression is significantly increased after a high dose of DETA-NO exposure in mESCs. Now, we aimed to investigate the role of EGR-1 in *Pdx1* expression at early β cell development. It has previously been described that EGR-1 is bound to regulatory areas III and IV, located at around 650 and 6000 base pairs (bp), respectively, upstream of the translation starting site of the *Pdx1* promoter in mouse and rat β cells (MIN6 and INS-1 cell lines, respectively). In this study, we used the JASPAR database [27] to elucidate the EGR-1 consensus binding sites in the *Pdx1* promoter. Analysis of 2000 bp of *Pdx1* promoter, upstream of its translation starting site, was carried out, and transcription factors with relevant roles in the control of pluripotency core and endodermal transcription factors were identified. Interestingly, the JASPAR database predicted with high confidence scores (total score 10.403 and relative score 0.8688), as not described before, as far as we know, an EGR-1 consensus binding site at 1064 to 1054 bp upstream of the translation starting site of the *Pdx1* promoter (TGCGGGGGGGG) (Figure 1A). Thus, we aimed to investigate if EGR-1 could bind to this novel site in mESCs and insulin-producing cells (INS-1E). We first verified that EGR-1 mRNA and protein levels do not vary in NO-exposed mESCs (Figure 1B,D,E), which indicates that the potential role of EGR-1 in mESC differentiation is not due to its differential expression after NO exposure. We also confirmed that cells cultured in the presence of leukemia inhibitory factor (LIF) (control condition) and treated with NO showed a significant increase in *Pdx1* expression, as reported in our previous studies [25,26] (Figure 1C,D,E). After these validations, chromatin immunoprecipitation (ChIP) studies were undertaken to study if EGR-1 binds to the *Pdx1* promoter at the novel position proposed by the JASPAR database. The results showed that EGR-1 binds to the *Pdx1* promoter in a control condition, and it is released after NO treatment (Figure 1F). To test, if EGR-1 also binds to the *Pdx1* promoter in the reported regulatory area III of insulin-producing cells (MIN6 and INS-1 cell lines), we performed ChIP assays in this same area in mESCs. The results showed that EGR-1 also binds to regulatory area III in mESCs, but no significant changes in EGR-1 binding to the *Pdx1* promoter after DETA-NO treatment were observed (Appendix A). As a control, the occupation of EGR-1 over the *Pdx1* promoter in INS-1E cells was performed, showing that EGR-1 is bound to regulatory area III of the *Pdx1* promoter in a functional β cell line (Appendix A).

### 2.2. EGR-1 Has a Repressor Role on Pdx1 Expression in mESCs

To elucidate the action of EGR-1 on *Pdx1* expression at an early stage of β cell development, EGR-1 loss and gain of function experiments were conducted in mESCs cultured under pluripotent conditions and DETA-NO treatment. The EGR-1 loss of function study was carried out by transfecting mESCs with a pool of short hairpin RNA (shRNA) targeting EGR-1 (shEGR1). This assay led to a decrease in EGR-1 expression of around 40% (Figure 2A), which allowed a significant increase in *Pdx1* expression (Figure 2B). This result suggests that EGR-1 acts as a repressor of *Pdx1* expression in an early stage of differentiation of mESCs. Conversely, EGR-1 overexpression (Figure 2C,D) did not affect *Pdx1* expression, neither under control nor DETA-NO conditions (Figure 2E,F). Thus, the repressor role of EGR-1 in *Pdx1* expression was not observed in the former assay. This result suggests that other mechanisms are involved in *Pdx1* regulation by EGR-1 in NO-exposed cells.

### 2.3. EGR-1 Binding to Pdx1 Promoter Is Dependent on DNA Methylation and Its Acetylation Status

Since binding of EGR-1 to its consensus sequences is reported to be affected by DNA methylation [28], and mESCs exposed to DETA-NO showed changes in DNA methyltransferases’ (Dnmts’) expression at RNA levels (Appendix A), we aimed to elucidate if *Pdx1* promoter methylation is affected by DETA-NO exposition in mESCs. Thus, *Pdx1* promoter methylation was analyzed by bisulfite sequencing PCR (BSP) and pyrosequencing. To this end, we studied, by BSP, the methylation levels of 11 CpG sites located around 1000 bp upstream of the translation starting site of the *Pdx1* promoter. BSP results showed that exposure to DETA-NO increased the methylation level of the CpG site of the EGR-1 binding site (CpG site referred to as 7), from 60% in cells cultured under pluripotent conditions to 90% after NO treatment (Figure 3A and Appendix A). Moreover, pyrosequencing of bisulfite-converted DNA was carried out to validate this result. It was confirmed that NO exposure increases methylation of the EGR-1 binding site and neighboring CpG sites. Specifically, bisulfite pyrosequencing results showed that DNA methylation of CpG site 7 increased from 59.37% to 80.88% after DETA-NO treatment (Figure 3B). Thus, it is proposed that the increase of DNA methylation level of the EGR-1 binding site CpG may interfere with the EGR-1 binding to the *Pdx1* promoter after NO treatment.

Furthermore, the repressor role of EGR-1 has been described to be modulated by P300/CBP-dependent acetylation [29,30]. Moreover, our previous study revealed that the acetyltransferase P300 is bound to the *Pdx1* promoter in mESCs cultured in control condition, and DETA-NO treatment decreases this binding. In addition, we demonstrated that treatment with a p300 inhibitor (P300i) induces significant *Pdx1* expression [26]. Thus, we set out to identify if EGR-1 could be implicated in this mechanism. For this purpose, cells were treated with a p300 inhibitor (P300i) and valproic acid (VPA), a histone deacetylase (HDAC) inhibitor, alone or in combination with DETA-NO, and the *Egr1* and *Pdx1* expression levels were tested. The results showed that the addition of p300i significantly reduces *Egr1* expression (*p* = 0.0023), while it significantly induces *Pdx1* (*p* = 0.0023). The combination of DETA-NO and p300i exhibited a cooperative effect on *Pdx1* expression (*p* = 0.0363 compared to DETA-NO condition). Conversely, VPA treatment significantly decreased *Pdx1* expression (*p* = 0.0021), but no expression changes were observed in cells exposed to DETA-NO (Figure 3C,D).

Finally, to elucidate if these changes in *Pdx1* expression were correlated with EGR-1 binding to its promoter, we analyzed the EGR-1 occupation on the *Pdx1* promoter by ChIP assays. The results showed that treatment with P300i tends to release EGR-1 from the *Pdx1* promoter, while a higher occupancy was observed in cells exposed to VPA in the presence of DETA-NO, compared to DETA-NO alone, although no significant changes were observed (Figure 3E). We also revealed that these changes are accompanied by epigenetic changes in the H3K27me3/H3K4me3 balance at the *Pdx1* promoter in the EGR-1 binding site region (Figure 3F).

These results indicate that the EGR-1 binding capacity to the *Pdx1* promoter is dependent on P300 activity, and DNA methylation of its consensus binding site may affect EGR-1 occupation, but other factors might be also involved in *Pdx1* regulation by the EGR-1 transcription factor (Figure 4).

## 3. Discussion

It has recently been reported that EGR-1 plays a crucial role in promoting β cell identity, maintaining β cell characteristics, and repressing non-β cell programs [19]. Moreover, EGR-1 is known to contribute to the glucose responsiveness and function of pancreatic β cells through the regulation of insulin and *Pdx1* expression [20,21]. *Pdx1* is a transcription factor essential for pancreatic development and β cell maturation, and appropriate regulation of its expression is instrumental to the generation of insulin-producing cells from PSCs. We previously defined that exposure to high concentrations of DETA-NO causes mESCs’ differentiation by downregulating pluripotency genes such as *Nanog* and *Oct4*, and upregulating differentiation genes including *Pdx1* [25]. We also described some of the mechanisms involved in *Pdx1* upregulation after DETA-NO treatment. Specifically, we revealed that PRC2 and P300 negatively regulate *Pdx1* gene expression in mESCs cultured under pluripotent conditions, and we observed that NO induces changes in the methylation pattern of the *Pdx1* promoter and leads to a shift in the balance of H3K27me3 and H3K4me3 occupancy at the proximal CpG island [26]. Thereafter, the present study was focused on elucidating the mechanisms by which EGR-1 regulates the expression of *Pdx1* during endoderm differentiation triggered by exposure to DETA-NO in mESCs.

The studies carried out by Eto et al. reported that EGR-1 binds to conserved regulatory area III [21], defined by Gerrish et al. [22] of the *Pdx1* promoter in the MIN6 cell line, describing EGR-1 as a positive regulator of this gene. Our results are in agreement with this finding as we reported EGR-1 occupancy in regulatory area III in INS-1E cells. Nevertheless, the present study has also identified a novel EGR-1 binding site located at −1064 to −1054 bp upstream of the translation starting site of the *Pdx1* promoter in mESCs. We have shown by EGR-1 loss and gain function assays that EGR-1 plays a repressor role in *Pdx1* expression during NO-induced differentiation of mESCs. In the same context, it has been reported that EGR-1 inhibition may be relevant for the differentiation of PSCs into insulin-producing cells [23]. In addition, Tsugata et al. recently described how the inhibition of EGR-1 in an early phase induces the differentiation of human PSCs into pancreatic endoderm and insulin-producing cells, but the inhibition of Egr-1 in a late phase suppresses the differentiation process [24]. Altogether, these results seem to indicate that EGR-1 could play a different role with the *Pdx1* promoter depending on the differentiation state.

This report also revealed some of the mechanisms by which EGR-1 regulates *Pdx1* in mESCs. We showed that EGR-1 occupies the *Pdx1* promoter in mESCs cultured under conditions that preserve pluripotency, and it releases the promoter after NO exposure. Our results suggest that these changes to EGR-1 binding on the *Pdx1* promoter are dependent on the level of CpG methylation status of the EGR-1 consensus binding site. The methylation studies shown here indicate that NO modifies the methylation pattern of the *Pdx1* promoter, specifically increasing the methylation of the EGR-1 binding site and neighboring CpG sites. These results are in agreement with our previous findings. We observed a significant increase in the methylation level of the proximal and distal CpG islands of the *Pdx1* promoter after NO treatment [26]. Moreover, several studies have previously reported that the methylation state of the EGR-1 binding site affects its ability to bind to DNA [31,32,33]. Additionally, the finding that pharmacological inhibition of the histone P300 acetyltransferase activity led to enhanced expression of *Pdx1* suggests that EGR-1 acetylation by P300 could play a role in this regulatory mechanism. Besides this, VPA exposure decreased *Pdx1* expression in PSCs. These results are consistent with our previous report showing that P300 is downregulated after DETA-NO exposure in mESCs, while *Pdx1* is negatively regulated by P300 [26]. Furthermore, cells exposed to p300i and VPA treatments showed slight changes in EGR-1 occupancy in the *Pdx1* promoter, which reinforces our proposal that the EGR-1 acetylation status may regulate its DNA binding capacity. However, the EGR-1/P300 interaction and EGR-1 acetylation status remain to be elucidated. These results also agree with the study by Liu et al., who described how P300 and other proteins are essential for EGR-1 activation and cell proliferation and survival [32].

Some studies have previously demonstrated a dual action of EGR-1. Yu et al. reported that EGR-1 binds directly to p300 regulatory sequences, and they form an acetylated EGR-1/P300/CBP complex activating growth and survival. They demonstrated that after an exogenous stimulus, EGR-1 is deacetylated and the p300/CBP promoter is repressed, leading to apoptosis [30]. Furthermore, EGR-1 has been described to act as a transcriptional repressor in melanoma cells and as an activator in breast, colon, and prostate cancer cells of the heparanase gene [31]. Our results also suggest that EGR-1 plays a dual role in *Pdx1* regulation during β cell development, playing a repressive role at an early stage and engaging in positive transcriptional activity at a later stage.

In conclusion, this study reveals a novel EGR-1 binding site on the *Pdx1* promoter and identifies EGR-1 as a repressor transcription factor of the *Pdx1* gene in mESCs. We have defined how EGR-1 occupancy in the *Pdx1* promoter depends on the methylation level of the EGR-1 binding site and histone P300 acetyltransferase activity. These results provide support for EGR-1 inhibition as a relevant target for efforts to direct PSCs into functional insulin-producing cells.

## 4. Materials and Methods

### 4.1. Cell Culture and Treatments

mESCs line R1/E (American Type Culture Collection (ATCC), University Boulevard, Manassas, VA, USA) was used. INS-1E cells were provided by Dr. P. Maechler (Geneva University, Geneva, Switzerland). R1/E cells were cultured on Nunclon surface gelatinized dishes and maintained at 37 °C with 5% CO_2_ in Dulbecco’s modified Eagle’s medium (DMEM) (Gibco, Carlsbad, CA, USA), supplemented with 15% heat-inactivated fetal bovine serum (FBS) (Hyclone, Logan, UT, USA), 0.1 mM β-mercaptoethanol (Gibco, Paisley, Scotland, UK), 2 mM L-glutamine (Gibco, Paisley, Scotland, UK), 1% minimum essential medium (MEM) nonessential amino acids (Gibco, Paisley, Scotland, UK), and 100 U/mL penicillin:100 μg/mL streptomycin (Gibco, Paisley, Scotland, UK). The undifferentiated state was maintained by adding 1000 U/mL LIF (ESGRO, Chemicon, Charlottesville, VA, USA). The R1/E cell culture protocol consisted of three days in the presence of LIF and 19 h with 500 μM DETA-NO (Sigma-Aldrich, St. Louis, MO, USA). Cells were then detached by trypsinization for 5 min and recovered by centrifugation. When appropriate, cells were exposed after three days of culture over 20 h to 50 μM C646 (P300i) (Millipore, Darmstadt, Germany) and 10 µM VPA (Sigma-Aldrich). INS-1E cells were cultured in Roswell Park Memorial Institute (RPMI)-1640 medium (Lonza, Verviers, Belgium), supplemented with 10% FBS, 10 mM N-2-hydroxyethyl piperazine-N-2-ethane sulfonic acid (HEPES; Gibco, Paisley, Scotland, UK), 1 mM sodium pyruvate (Gibco, Paisley, Scotland, UK), 100 U/mL penicillin:100 μg/mL streptomycin, 2 mM L-glutamine, and 50 μM β-mercaptoethanol (Gibco, Paisley, Scotland, UK).

### 4.2. RNA Isolation, Reverse Transcription, PCR, and Real-Time PCR Analysis

Total RNA was extracted using Easy Blue^®^ reagent (Intron Biotechnology, Gyeonggi-do, South Korea) and following the chloroform/isopropanol purification procedure. cDNA synthesis was performed with 1 μg total RNA using Moloney murine leukemia virus reverse transcriptase (M-MVL RT) (Promega, Madison, WI, USA) and random primers according to the manufacturer’s instructions. For real-time PCR analysis, endogenous mRNA levels were measured based on SYBR Green (Applied Biosystems, Foster City, CA, USA) detection with an ABI Prism 7500 machine (Applied Biosystems). Results were normalized with the β-actin expression. The real-time PCR primers used are shown in Appendix A.

### 4.3. Protein Extraction and Western Blotting

Cells were trypsinized, centrifuged at 230× *g*, and washed once with cold phosphate-buffered saline (PBS). Cell pellets were then resuspended and incubated in radioimmunoprecipitation assay (RIPA) buffer (Sigma-Aldrich, St. Louis, MO, USA), supplemented with a protease inhibitor (Sigma-Aldrich) and phosphatase inhibitor cocktail (Sigma-Aldrich) for 45 min on ice, and sonicated with four pulses (10 s each) at 10% amplitude on an ultrasonicator (Branson Ultrasonics Corporation, Danbury, CT, USA). After centrifugation at 9300× *g*, the protein content in supernatants was quantified by a Bradford assay. Proteins were denatured in Laemmli buffer supplemented with 2.5% β-mercaptoethanol (Sigma-Aldrich) for 10 min at 96 °C. Proteins were separated using sodium dodecyl sulfate-polyacrylamide gel electrophoresis (SDS-PAGE) and transferred to polyvinylidene difluoride (PVDF) membranes (GE Healthcare, Buckinghamshire, UK). Membranes were then blocked with PBS 5% skim milk (Becton and Dickinson, Franklin Lakes, NJ, USA) for 1 h at room temperature (RT). Then, membranes were probed with anti-EGR1 (1:500, Santa Cruz Biotech), dissolved in 4% BSA overnight at 4 °C, and anti-*Pdx1* (1:1000, Abcam) in Tris-buffered saline (TBS) (Sigma-Aldrich) supplemented with 0.1% Tween 20 (TBS-T) containing 5% skim milk, overnight at 4 °C. Membranes were incubated with anti-β-actin (1:10000, Sigma-Aldrich) on TBS-T supplemented with 5% skim milk for 1h at RT, and washed three times for 5 min with TBS-T. The secondary antibodies used were anti-rabbit IgG (1:20000, Sigma-Aldrich) and anti-mouse IgG (1:40000, Jackson ImmunoResearch, Suffolk, UK), and proteins were detected by chemiluminescence.

### 4.4. Bisulfite Sequencing PCR (BSP)

A region of approximately 2000 base pairs (bp) of the *Pdx1* promoter was analyzed with the software Methyl Primer Express v1.0. (Applied Biosystems) to identify CpG-rich regions. Primers designed for these regions are listed in Appendix A. Then, genomic DNA from 7.5 × 10^4^ cells was converted with sodium bisulfite using a Cells-to-CpGTM Bisulphite Conversion Kit (Applied Biosystems, Waltham, MA, USA). Converted DNA was amplified by PCR using MyTaqTM HS Red DNA Polymerase, and then PCR products were purified and cloned into the pGEM-T vector to obtain *Escherichia coli* (*E. coli*) colonies. Ten colonies per condition were analyzed by PCR and later sequenced in a DNA Analyzer 3730 (Applied Biosystems). As the technical control, we treated the isolated genomic DNA of mESC cultured under a control condition with the CpG methyltransferase, M.SssI (New England BioLabs, Ipswich, MA, USA) according to the manufacturer’s instructions. The methylation of the proximal CpG island of the *Pdx1* promoter was studied. The results were analyzed by BiQ Analyzer Software (Max-Planck-Institute and Saarland University, Saarbrücken, Germany).

### 4.5. Bisulfite Pyrosequencing

*Pdx1* promoter methylation status results obtained by BSP were confirmed by pyrosequencing. Sodium bisulfite modification of genomic DNA of 7.5 × 10^4^ cells was carried out as described above. Converted DNA was eluted in 15 μL and 2 μL for each PCR cycle. The primers used for PCR and sequencing were designed using PyroMark assay design software, version 2.0.01.15 (Qiagen, Hilden, Germany). The pyrosequencing primers are shown in Appendix A. These primers were designed to hybridize with CpG-free sites to ensure methylation-independent amplification. PCR was performed with biotinylated primers to convert the PCR product to single-stranded DNA templates, using the Vacuum Prep Tool (Biotage, Uppsala, Sweden), according to the manufacturer’s instructions. Pyrosequencing reactions and methylation quantification were performed using a PyroMark Q24 System, version 2.0.6 (Qiagen).

### 4.6. Chromatin Immunoprecipitation Assay

Cells were crosslinked with 1% (*w*/*v*) formaldehyde (Sigma-Aldrich) for 10 min at 37 °C. Then, 3 × 106 cells were resuspended in lysis buffer containing 10mM NaCl, 10mM Tris HCl (pH 8), 3mM Cl2Mg, and 0.5 mM 1,4-dithiothreitol (DTT) (Sigma-Aldrich) supplemented with protease inhibitors (Sigma-Aldrich), for 10 min on ice. Cells were then centrifuged for 5 min at 800× *g* and 4 °C. Supernatants were discarded and the nuclei-containing fraction was washed by inversion with buffer containing 10 mM Tris HCl (pH 8), 15 mM NaCl, and 60 mM KCl. Cell nuclei were centrifuged for 5 min at 800× *g* and 4 °C and incubated with washing buffer supplemented with 3 mM CaCl2, protease inhibitors, 0.5 mM DTT, and 10 μL micrococcal nuclease (1:200 dilution) (New England BioLabs, Ipswich, MA, USA), and they were then incubated for 20 min at 37 °C with orbital shaking. The nuclease activity was halted by adding 20μL of 0.5 mM ethylenediaminetetraacetic acid (EDTA). Cell nuclei were centrifuged for 5 min at 800× *g* and 4 °C and lysed with buffer containing 150 mM NaCl, 50 mM Tris HCl (pH 7.5), 5 mM EDTA, 0.5% NP-40, 1% Triton X-100, and 0.01% SDS, and they were sonicated with three pulses for 10 sec each at 10% amplitude in a Branson sonifier. Extracts were then centrifuged for 10 min at 9300× *g* and 4 °C. Supernatants containing chromatin with an average size of 500 bp were immunoprecipitated with anti-EGR1 (3 μg, Santa Cruz Biotechnology, Dallas, TX, USA), anti-H3K4me3 (1 μg; Abcam), anti-H3K27me3 (1 μg; Cell Signaling Technology Inc.; Danvers, MA, USA), and mouse IgG1 isotype control (3μg, Cell Signaling), used as control. Next, 15 μL of Dynabeads^®^ (Invitrogen, Dynal AS, Oslo, Norway) was used to prepare Ab-bead complexes, incubated for 30 min at 4 °C under rotation on dilution buffer (0.01% SDS, 1.1% Triton X-100, 1.2 mM EDTA, 16.7 mM Tris HCl pH8.1, and 167 mM NaCl). Then, chromatin was added and incubated for 30 min under the same conditions; 3% of chromatin was reserved as the input control. Washes of the complex were carried out once with low-salt buffer (0.1% SDS, 1% Triton X-100, 2 mM EDTA, 20 mM Tris HCl pH 8.1, 150 mM NaCl), once with high-salt buffer (0.1% SDS, 1% Triton X-100, 2 mM EDTA, 20 mM Tris HCl pH 8.1, 500 mM NaCl), once with LiCl buffer (0.25 M LiCl, 1% NP40, 1% deoxycholate, 1 mM EDTA, 10 mM Tris HCl pH 8), twice with TE buffer (10 mM Tris HCl pH 8.1, 1 mM EDTA), and finally, eluted with 500 μL elution buffer (1% SDS, 0.1 M NaHCO_3_). DNA was purified by a phenol/chloroform procedure. ChIP analysis was performed by real-time PCR using SYBR Green (Bio-Rad, Hercules, CA, USA). The promoter occupancy was determined by the percentage input method relativized to the β-actin expression of the input samples. To this end, 3% inputs were employed in the assays, and an adjusted Ct for the inputs was calculated accordingly. Percentage input data were normalized to IgG binding in each condition, so IgGs are represented as 1. The primers used are listed in Appendix A.

### 4.7. Egr-1 Loss and Gain of Function

In loss of function experiments, R1/E cells were transfected after two days in culture with a pool of short hairpin RNA (shRNA) (Sigma-Aldrich), while in gain of function experiments, EGR-1 overexpression was achieved by transfecting R1/E cells with the plasmid pCG-HA-EGR-1, kindly provided by Dr. Shigeru Taketani. Cells were transfected using Fugene HD transfection reagent (Promega, Madison, WI, USA). In both cases, before the transfection, a Fugene:DNA complex was created with a ratio of 4:1, incubated for 1 h at RT in Opti-MEM medium (Gibco, Waltham, MA, USA). Then, the complex was added to the cells, covering the plate with Opti-MEM medium, then incubating for 3 h at 37 °C and 5% CO_2_. Finally, the plate’s volume was completed with the corresponding culture medium. After leaving overnight, the medium was changed and DETA-NO was added, and after 19 h, the cells were collected. R1/E cells transfected with the shEGR1 pool were selected with 2 µg/mL of puromycin 48 h after the transfection. The degree of silencing and overexpression was assessed by real-time PCR. The primers used are listed in Appendix A.

### 4.8. Statistical Analyses

The data represented are the means ± standard deviation (SD) of at least three independent experiments (unless specified otherwise in the figure caption). Comparisons between values were analyzed using Student’s *t*-test with GraphPad Prism v7 (GraphPad Software, San Diego, CA, USA). *p*-values < 0.05 were considered statistically significant.

## Figures and Tables

**Figure 1 ijms-23-03920-f001:**
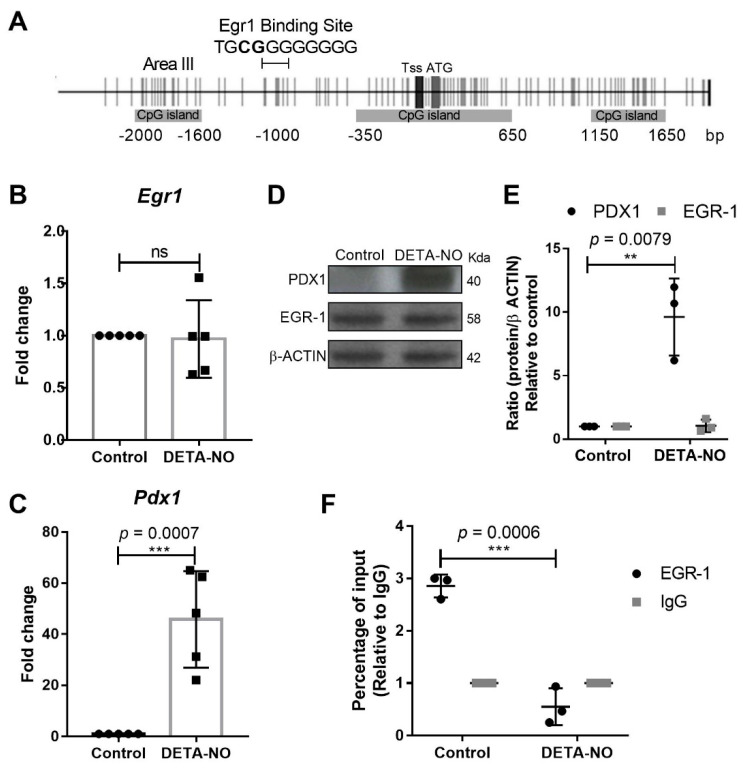
EGR-1 binds to *Pdx1* promoter in mESCs. (**A**) *Pdx1* promoter scheme shows CpG sites (vertical gray lines), CpG islands (horizontal gray rectangles), transcription start site (TSS) and translation start site (ATG) represented by vertical gray rectangles, and EGR-1 consensus sequence (TGCGGGGGGGG) on *Pdx1* according to the JASPAR database and regulatory area III (Area III). Analysis of (**B**) *Egr-1* and (**C**) *Pdx1* expression after DETA-NO treatment by real time-PCR. These values were normalized to expression values of β-actin, used as the loading control and analyzed using the ΔΔCt algorithm. They represent the average of five independent experiments. Data are means ± standard deviation (SD). (**D**) PDX1 and EGR-1 expression were analyzed by Western blotting in control and DETA-NO conditions. β-actin was used as loading control. The image shown is the most representative of three independent experiments. (**E**) Western blotting quantification of three independent experiments of PDX1 and EGR-1 proteins, relativized to β-actin expression, using ImageJ software. (**F**) Chromatin immunoprecipitation (ChIP) assays of EGR-1 on *Pdx1* promoter at JASPAR database-predicted location in mESCs. Result shows the means ± SD of three independent experiments. Y-axis corresponds to the percentage input relativized to IgG binding. Data with ** *p* < 0.01 or *** *p* < 0.001 were considered statistically significant.

**Figure 2 ijms-23-03920-f002:**
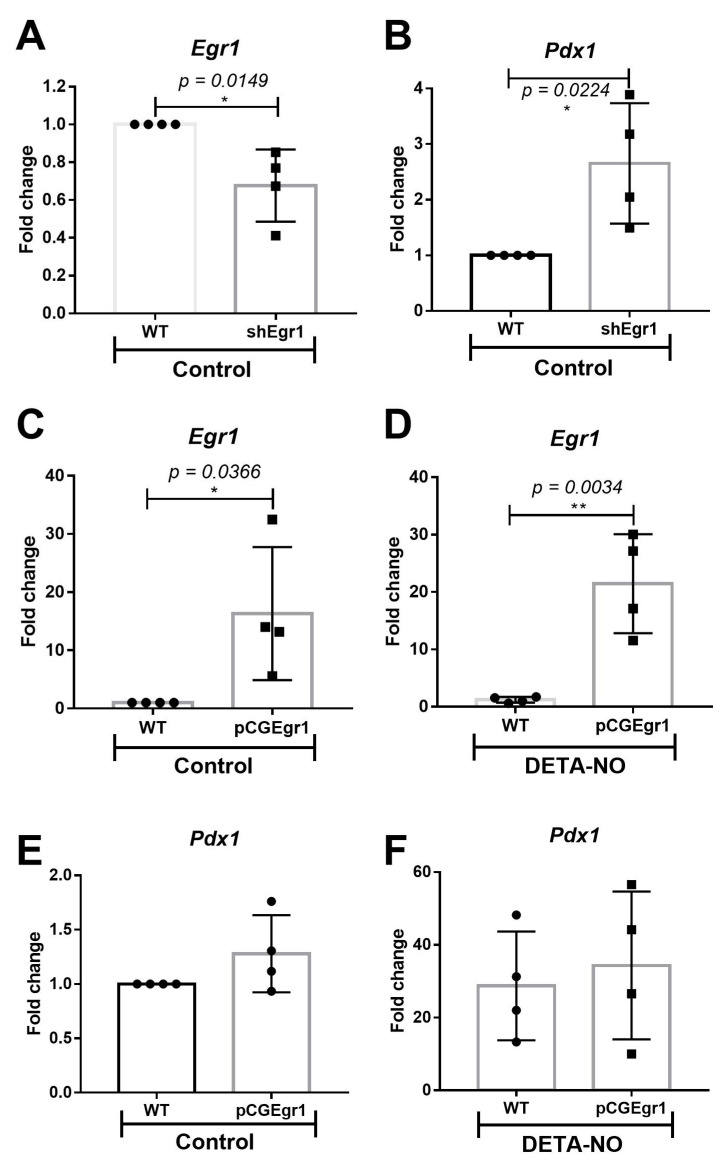
Role of EGR-1 in *Pdx1* expression. (**A**,**B**) EGR-1 loss of function assays. mESCs were transfected with a shEgr1 pool and selected with puromycin. (**A**) *Egr-1* and (**B**) *Pdx1* expression were analyzed by real time-PCR in wild-type (WT) cells and silenced cells (shEgr1) cultured in control condition. (**C**–**F**) EGR-1 gain of function assays. mESCs were transfected temporally with the vector pCG-Egr1-HA. (**C**,**E**) EGR-1 and (**D**,**F**) *Pdx1* expression were analyzed by real time-PCR in WT and EGR-1-overexpressing cells (pCGEgr1) cultured in control and DETA-NO conditions. Data were normalized to the expression values of β-actin, used as the loading control and analyzed using the ΔΔCt algorithm. Bar graphs represent the means ± SD of four independent experiments. Data with * *p* < 0.05 or ** *p* < 0.01 were considered statistically significant.

**Figure 3 ijms-23-03920-f003:**
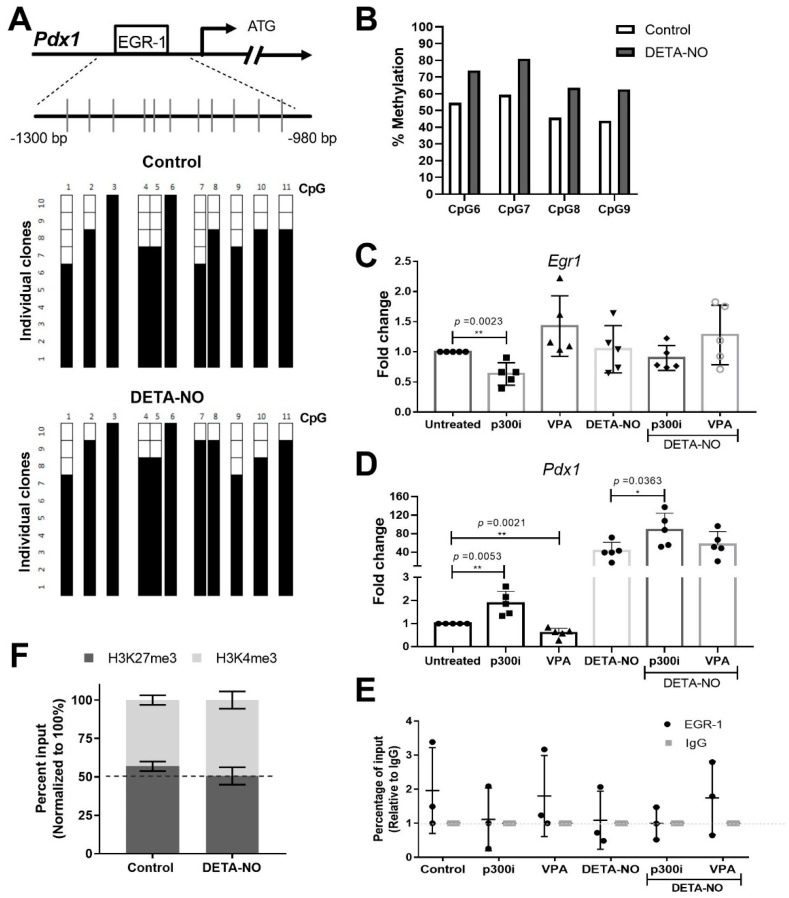
EGR-1 binding to *Pdx1* promoter is dependent on DNA methylation and histoneP300 acetyltransferase activity. (**A**,**B**) Analysis of *Pdx1* promoter methylation status. a) Schematic depiction of the *Pdx1* promoter. Short vertical lines represent the 11 CpG dinucleotides studied. Results are shown of bisulfite PCR sequencing of 10 individual clones in control and DETA-NO conditions. Presence of a methylated (black square) or unmethylated (white square) cytosine is indicated. (**B**) Specific CpG site methylation analysis of EGR-1bs (CpG referred to as 7) and neighboring CpG sites on *Pdx1* promoter, analyzed by pyrosequencing. (**C**) EGR-1 and (**D**) *Pdx1* expression by real-time PCR in control cells and cells treated with DETA-NO, p300i, and valproic acid (VPA), alone or in combinations. Graph shows the average ± SD of five independent experiments. (**E**) EGR-1 binding analysis of *Pdx1* promoter by chromatin immunoprecipitation (ChIP) assay. Graph shows the average ± SD of three independent experiments. Y-axis corresponds to the percentage input relative to IgG. (**F**) Stacked bars for bivalent mark dynamics of H3K27me3 and H3K4me3 analyzed by ChIP assays. They represent an average of two independent experiments. Data with * *p* < 0.05 or ** *p* < 0.01 were considered statistically significant.

**Figure 4 ijms-23-03920-f004:**
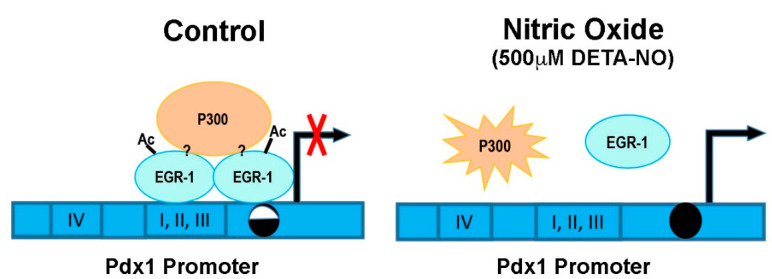
Overview of *Pdx1* regulation by EGR-1 and P300 in mESCs. This graph represents the *Pdx1* promoter in control and DETA-NO conditions. Control condition: mESCs cultured in medium supplemented with LIF. DETA-NO condition: mESCs cultured in medium supplemented with LIF and exposed to 500 µM DETA-NO for 19 h. I, II, III, and IV represent the conserved regulatory areas in the *Pdx1* promoter. The circles represent the CpG site of the EGR-1 binding site studied on the *Pdx1* promoter. Methylation grade is shown by the circle color (white and black: 40–60%; black: 80–100%). Question marks (?) symbolize that EGR-1/P300 interaction and the role of other proteins in regulating *Pdx1* expression remain to be elucidated. Ac (acetylation); diethylenetriamine nitric oxide adduct (DETA-NO); early growth response-1 (EGR-1); pancreatic and duodenal homeobox 1 (*Pdx1*).

## Data Availability

Not applicable.

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
