# Peer review of "Pdx1* Is Transcriptionally Regulated by EGR-1 during Nitric Oxide-Induced Endoderm Differentiation of Mouse Embryonic Stem Cells"

_ijms, 2022, doi:10.3390/ijms23073920_

Round 1
Reviewer 1 Report
The authors have revised and improved the manuscript significantly. This may be accepted for publication.
Reviewer 2 Report
The authors adressed all my comments.
This manuscript is a resubmission of an earlier submission. The following is a list of the peer review reports and author responses from that submission.
Round 1
Reviewer 1 Report
The manuscript by Salguero-Aranda et al. describes the role of EGR-1 in regulating pdx1 expression in mESCs. The main discovery show that EGR-1 has a dual effect in maintaining pdx1 expression based on the embryonic stage of development. During differentiation EGR-1 shows a repressive effect on pdx1 expression, while in mature beta-cell it has a positive effect on pdx1 expression. These results are of high relevance for the field of beta-cell differentiation in vitro and shed lights on potential new avenue to promote efficient differentiation of mESCs towards beta-cells
While the experimental approaches look satisfying there are many aspects that should addressed:
- The full paragraph from line 51 to line 69 is confusing and does not add any information on the specific topic of the paper. It just shows that EGR-1 has pleiotropic effects. This could be easily stated with a shorter sentence.
- The argument introduced in line 70 is unclear. What the authors refer to with “…as a critical factor in the development of pancreatic islet failure…compensatory response in pancreas beta-cells”. Perhaps a reference could clarify the point.
- The paragraph from line 108 to line 114 should be rewritten. The authors should put more emphasis on the fact that they identified a novel binding site of EGR-1 in the mESCs.
- The sentence in line 190 should be rephased. “Synergistic effect” is a very specific definition that requires supporting experimental data. Considering the low statistical significance it would be more appropriate to consider the combined effect of DETA-NO and P300i as cooperative.
Overall, the manuscript has some interesting outcomes. The data reported are convincing with beneficial effect for the Diabetes field.
I would recommend publication after revisions
Reviewer 2 Report
In this manuscript, Aranda and coauthors studied the regulation of Pdx1 by EGR-1 during nitric ox-2 ide-induced endoderm differentiation. The authors show that treatment of mESCs with Nitric Oxide (NO) induces the expression of Pdx1. They also suggested that Pdx1 expression is associated with PRC2 and P300 release from Pdx1 promoter that leads to changes in DNA methylation. Together, the authors suggested that Egr-1 regulates mESC differentiation through modulating Pdx1 transcription.
Although this article is interesting, I have a few comments to improve the manuscript.
- Although the EGR-1 chip assay suggests the association with Pdx1 promoter, the difference in binding upon DETA-NO treatment is not very clear. The qPCR analysis should be normalized with GAPDH/18S gDNA. In addition, Normalization with input may not show the actual difference in pulldown in the two conditions mentioned Figure 1. The IgG should be represented as 1 and the EGR-1 pulldown should be shown as fold enrichment compared to IgG.
- Figure 1 E & 1F are from 2 biological replicates. Again, this experiment was performed using input as control. Please perform a few more repeats for this experiment and use some internal controls for normalization.
- Figure 3B,D; The change in methylation should be repeated and shown with error bar. In panel D, it says 5-9 repeats. It might be better to have equal number of experiments in each panel.
- Figure 3C. Please show a loading control for western blot analysis.
- Figure 3F, The legend says it’s a repeat of 2 experiments while the data points shows many data sets. Please correct it accordingly.